# Optimization of Green Vehicle Paths Considering the Impact of Carbon Emissions: A Case Study of Municipal Solid Waste Collection and Transportation

Tingting Li [1], Shejun Deng [1,*], Caoye Lu [2], Yong Wang [3] and Huajun Liao [4]

1   College of Architectural Science and Engineering, Yangzhou University, Yangzhou 225009, China; litt370@163.com
2   Jiangsu Communications Planning and Design Institute Limited by Share Ltd., Nantong 226000, China; lucaoye1996@163.com
3   School of Economics and Management, Chongqing Jiaotong University, Chongqing 400074, China; yongwx6@gmail.com
4   Chengdu Map Wisdom Technology Co., Ltd., Chengdu 610041, China; fanglingying@supermap.com
*   Correspondence: yzrx6@163.com; Tel.: +86-13852727958

**Abstract:** In recent years, the waste produced as a result of the production and consumption activities of urban residents has led to significant environmental degradation and resource wastage. This paper focuses on the research object of municipal solid waste (MSW) collection and transportation based on the concept of "sustainable development and green economy". Firstly, this study examines the current state of urban domestic garbage collection and transportation. It analyzes the following challenges and deficiencies of the existing collection and transportation system: (1) the operating efficiency of garbage collection vehicles is low, resulting in a significant accumulation of waste on the roadside and within the community; (2) the vehicle collection and transportation routes are fixed, and there are empty vehicles running; (3) the amount of garbage on a route exceeds the vehicle's loading capacity, which requires the vehicle to perform a second round of collection and transportation. To enhance the efficiency of urban garbage collection and transportation and minimize the collection and transportation costs, we are investigating the problem of optimizing the path for green vehicles. To comprehensively optimize the fixed cost, variable cost, and carbon emission cost incurred during vehicle operation, a vehicle routing model with time windows is established, taking into account vehicle load constraints. Carbon emission coefficient and carbon tax parameters are introduced into the model and the "fuel-carbon emission" conversion method is used to measure the carbon cost of enterprises. An improved ant colony optimization (ACO) method is proposed: (1) the introduction of a vehicle load factor improves the ant state transfer method; (2) the updated pheromone method is improved, and additional pheromone is added to both the feasible path and the path with the minimum objective function; (3) the max–min ACO algorithm is introduced to address the issue of premature convergence of the algorithm; (4) the embedding of a 2-opt algorithm further prevents the ACO algorithm from falling into the local optimum. Finally, the calculation results based on the example data demonstrate that the algorithm has a significant advantage over the genetic algorithm (GA) and particle swarm optimization (PSO) algorithm. The total transportation distance determined by this algorithm is shorter than that of the GA and PSO methods, and the total cost of the scheme is 1.66% and 1.89% lower than that determined by GA and PSO, respectively. Compared to the data from the actual case, the number of vehicles required in the operation of this algorithm and model is reduced by three. Additionally, the total cost, fixed cost, and carbon emission cost incurred by the vehicles during operation were reduced by 31.2%, 60%, and 25.3% respectively. The results of this study help the station to collect and distribute waste efficiently, while also achieving the goals of energy saving, consumption reduction, and emission reduction.

**Keywords:** urban traffic; municipal solid waste management; vehicle route optimization; network analysis; green and low-carbon



## 1. Introduction

The rapid and constant growth of populations and increase in urbanization has led to a sharp increase in municipal solid waste (MSW) generation, which seriously affects the social economy and environment. The management of MSW primarily encompasses five key components: production, delivery, collection, transportation, and recycling. Collection and transportation play crucial roles in connecting the initial drop-off point of waste to the final treatment process. During the collection and transport process, we often need to consider many influencing factors, such as the social factor of residents' involvement in waste disposal and the method used for waste collection and transportation. At present, the modes of MSW collection and transportation can be categorized based on the following five factors: (1) whether the time of collection and transportation schedule is fixed, i.e., whether the vehicle collects and transports the waste in the city during a fixed period; (2) whether the location of the waste collection point is fixed, i.e., whether the node of the waste that the vehicle collects and transports every day is fixed; if it is fixed, then the phenomenon of waste piling up can occur, causing collection and transportation to run off-schedule; (3) whether it is classified for collection and transportation; (4) whether the vehicle uses an independent collection or joint collection mode; (5) and whether there is a transfer station; if there is a transfer station, the vehicle will transport the waste to the transfer station, and if not, the vehicle will transport the waste directly to the disposal site. Collection and transportation are considered high-cost components, comprising approximately 65–80% of the overall cost of waste management [1]. For many years, researchers have been concerned about the vehicle routing problem (VRP) of waste collection. It has been demonstrated through empirical evidence that the implementation of an efficient waste collection path can result in a substantial reduction in costs. Nevertheless, in light of the advancements in green and low-carbon cities, some researchers have proposed the green vehicle routing problem (GVRP) to reduce carbon emissions of the path.

The arc routing problem (ARP) can also be utilized for waste collection routing. Ghiani et al. [2] conducted a study for MSW collection and built an ARP to minimize the total distance traveled by the vehicles by first assigning arcs/edges to vehicles through clustering and then matching routes for the vehicles. The experimental results showed that the proposed system enables individuals to avoid overtime and reduces the total cost of the vehicles by 10%. Willemse and Joubert [3] researched the mixed capacity arc routing problem under time restrictions with intermediate facilities. Their research aimed to identify the best-performing constructive heuristic in terms of computational time and ability to find the least costly solution and smallest fleet size. The method that they employed to evaluate randomized heuristics is also of value to future studies on CARPs. In 2019, Willemse and Joubert [4] extended upon the constructive heuristic research conducted on the mixed capacitor arc routing problem under intermediate facility time constraints (MCARPTIF) from the research published in 2016 [3]. More advanced local search acceleration mechanisms from the literature were adapted and combined with the MCARPTIF and tested on the same instances. The execution time of the algorithm was decreased; however, the local search yielded poorer solutions. Lu et al. [5] focused on formulating and solving rich arc routing problems (RARPs) in city logistics in a congested urban environment. The authors proposed an analytical methodology that utilized a fluid queue model to calibrate link travel time. The calibration was achieved through the use of a polynomial arrival rate function. The systemic (social) impacts of vehicle routes were derived analytically and incorporated into an RARP model, where operational costs and social impacts were systematically considered in the design of route strategies and compared with examples at three different scales. Ghiani et al. [6] employed a new ant colony optimization procedure for the arc routing problem with intermediate facilities under capacity and length restrictions (CLARPIF), and the computational results showed substantial improvements. Mourão and Amado [7] proposed a new heuristic to generate feasible solutions for the extended CARP on hybrid graphs by constructing Eulerian and directed networks to generate feasible vehicle journeys based on selected maximal



circuits. Nie et al. [8] proposed an optimal configuration and arc routing problem (ARP) for sanitation vehicles operating on urban roads under multiple constraints. The physical road network was expanded into a spatio-temporal network, and the spatio-temporal trajectories of vehicles on the road network were portrayed. An optimal configuration and path planning model for sanitation vehicles was constructed, and a branch pricing algorithm was devised to effectively solve the model with precision.

This paper focuses on the path optimization problem for sanitation vehicles. At present, the research on VRP has been quite rich. It has been applied to various fields, and various VRP variants have been gradually developed, such as the capacitated vehicle routing problem (CVRP), vehicle routing problem with time window (VRPTW), green vehicle routing problem (GVRP), and multi distribution center vehicle routing problem (MDCVRP). Due to the restricted load capacity of vehicles, it is imperative to take into account capacity limitations when solving the VRP [9–16]. In the process of waste collection and transportation, in order to complete the work on time and efficiently, it is necessary to consider the constraints of the time window when optimizing the route [17–19]. The majority of waste trucks in the city are powered by fuel. In order to mitigate the carbon emissions associated with transportation, one potential solution is to optimize the route. Jabir, Panicker, and Sridharan [20] first to addressed the integration of carbon dioxide ($CO_2$) emissions into the vehicle routing problem. Their proposed model aimed to resolve the trade-off between cost and emission reduction, resulting in a substantial decrease in the total cost. Ziaei and Jabbarzadeh [21] considered the impact of carbon emissions on a multi-modal transport network for hazardous materials. Sherif et al. [22] incorporated the cost of carbon emissions into the objective function and built a multi-depot heterogeneous green vehicle routing optimization model for the battery supply chain network. Madden et al. [23] built a model to estimate carbon emissions from curbside organic waste collection based on waste collection route data, which showed that curbside collection was the largest contributor to overall transport emissions. Guo, Qian, et al. [24] proposed a three-dimensional ant colony optimization algorithm (TDACO) to solve the multi-compartment vehicle routing problem (MCVRP) in industries such as waste collection and incorporated carbon emissions into the state transition rules in the TDACO. Dayanara, Arvitrida, and Siswanto [25] constructed a vehicle routing optimization model with the number of waste collections, time windows, and carbon emissions as constraints. Liu and Liao [26] considered different types of vehicles working together for waste collection and built an optimization model to minimize economic costs and carbon emissions. Li et al. [27] comprehensively considered the fixed vehicle costs, early and delayed penalty costs, fuel costs, and the impacts of vehicle speed, load, and road gradient on fuel consumption and developed a hybrid genetic algorithm solution with variable neighborhood search. Zhou, Li, and Wang [28] took into account how vehicle load affects carbon emissions and constructed a model that they then tested for robustness to find the shortest route and reduce carbon emissions. Wang and Shan [29] established a multi-objective waste collection model combining transport distance, fuel consumption, and carbon emission and demonstrated the effectiveness and practicality of the algorithm. Lu [30] constructed a mathematical model with the optimization objective of minimizing economic cost and carbon emission cost to meet the low-carbon demand for waste collection and transportation. Li, Song, and Guo [31] established a cold chain logistics multi-temperature co-distribution path optimization model consisting of transportation cost, carbon emission cost, refrigeration cost, and loss cost with the lowest total cost as the objective function to achieve low-carbon collection and transportation. Martyushev et al. [32] studied the operational performance of electric vehicles and developed a simulation model to determine the range of an electric vehicle by cycles of movement. The effects of operating speed, drag, and mechanical forces on the operation of electric vehicles were considered in the modeling process.

Considerable research has been conducted on the mathematical models [33] pertaining to the VRP, as well as the algorithms developed for solving it. The algorithms utilized for solving the VRP are mainly divided into two categories: exact algorithms and heuristic

algorithms. The most widely used algorithm is the meta-heuristic algorithm, which mainly includes the genetic algorithm, ant colony algorithm, simulated annealing algorithm, etc. To minimize the time taken during the collection of bio-medical waste (BMW), Mohamed et al. [34] built an optimized vehicle route model for a set of six dedicated vehicles using a particle swarm optimization (PSO) algorithm. Moazzeni, Tavana, and Mostafayi Darmian [35] used a GA and grey wolf optimizer (GWO) to solve the dynamic location-arc routing optimization model for electric waste collection vehicles. Liu and He [36] designed a clustering ant colony algorithm to solve the optimization of domestic waste collection and transportation routes in Chengdu. Wang [37] formulated a multi-objective optimization model with time windows and used a particle swarm algorithm to solve it to obtain a dispatching scheme for collection vehicles under different modes. Zhao, Ma, and Liu [38] established a mathematical model to minimize transportation costs and vehicle fixed costs and proposed an improved ant colony algorithm to obtain a suboptimal solution. Experimental results showed the correctness of the proposed model and the effectiveness and optimization ability of the algorithm.

Although scholars have made a lot of achievements, there are still some limitations that have yet to be addressed. The impact of vehicle fixed costs, transportation costs, and carbon emission costs, as well as vehicle capacity and time window constraints on transportation costs, are rarely considered. For the solution algorithm for this problem, ACO has strong robustness but still has the weakness of easily falling into a local optimum.

Carbon peaks and carbon neutrality have emerged as highly significant subjects of discussion in the current discourse. China boasts the largest carbon trading system, which not only facilitates the reduction of carbon emissions but also contributes to the growth of GDP. The carbon market introduces a level of flexibility to the industry, enabling companies to strategically manage their $CO_2$ emissions in order to minimize costs. The aim of this paper is to reduce business costs and increase environmental benefits by optimizing the routes of sanitation vehicles. This study is based on real case data from Huzhou, Zhejiang, China. Firstly, the waste collection network in the study area is divided into five sub-areas by investigating the current situation of MSW collection and transportation, analyzing the distribution of waste nodes, and studying the waste generation pattern. A model is established for vehicle matching and path optimization with time window constraints and capacity constraints. An improved ant colony algorithm is designed to solve the model and compare the results with those of the other two algorithms.

The paper is structured as follows:

(1) Section 1: Introduction. The introduction provides the background and purpose of the study and summarizes the relevant research literature in related fields.

(2) Section 2: Model Formulation. This section introduces the research data, research questions, assumptions, and constraints and establishes a model for the matching and path optimization of MSW collection vehicles with time window constraints and capacity constraints.

(3) Section 3: Methodology. An improved ant colony algorithm is designed to solve the mathematical model, and the steps and contents of the algorithm are discussed in detail.

(4) Section 4: Results. The effectiveness of the improved ACO algorithm is verified by example data. By comparing the experimental results of PSO and GA, the algorithm designed in this paper has significant superiority.

(5) Section 5: Conclusion. This section summarizes the research findings and the progress made in this paper. It also summarizes the limitations of the research in this paper and explores the possibilities and improvement directions for future research.

## 2. Model Formulation

### 2.1. Research Data

The data for the study described in this paper were primarily obtained from waste collection data from Wuxing District, Huzhou City, Zhejiang Province, China.

In this paper, data on domestic waste nodes for 63 neighborhoods in the area and waste volume data generated for three consecutive months were obtained, as shown in Table 1.

**Table 1.** Daily average of domestic waste in each neighborhood.

| Block Number | Daily Average (kg) | Block Number | Daily Average (kg) |
|:---:|:---:|:---:|:---:|
| 1 | 16.2874 | 15 | 25.19309 |
| 2 | 38.20596 | 16 | 43.59313 |
| 3 | 116.1029 | 17 | 7.240346 |
| 4 | 82.4291 | 18 | 36.31804 |
| 5 | 27.22089 | 19 | 110.6039 |
| 6 | 125.5312 | 20 | 297.0145 |
| 7 | 43.59113 | 21 | 57.20373 |
| 8 | 28.9518 | 22 | 569.5574 |
| 9 | 152.7669 | 23 | 177.8744 |
| 10 | 69.16793 | 24 | 274.5275 |
| 11 | 27.14389 | 25 | 381.2409 |
| 12 | 52.62627 | 26 | 1412.876 |
| 13 | 51.00258 | 27 | 844.055 |
| 14 | 52.5404 | 28 | 28.06824 |

Data from 98 waste collection nodes and 8 sanitation vehicle operations in the region were analyzed. The data mainly consist of GPS trajectory data from vehicles, which include 65,499 entries. The data include latitude and longitude coordinates, position, speed, time, and operation status, as shown in Table 2.

**Table 2.** Main information from sanitation vehicle data.

| Longitude | Latitude | Speed (km/h) | Time | Status |
|:---:|:---:|:---:|:---:|:---:|
| 120.2348 | 30.84301 | 27 | 4:00 | In operation |
| 120.2348 | 30.84527 | 30 | 4:00 | In operation |
| 120.2349 | 30.84772 | 18 | 4:01 | In operation |
| 120.2349 | 30.84777 | 18 | 4:01 | In operation |
| 120.2351 | 30.84779 | 20 | 4:01 | In operation |
| 120.2418 | 30.84807 | 28 | 4:02 | In operation |
| 120.2447 | 30.84797 | 0 | 4:03 | Stalled |
| 120.2447 | 30.84797 | 0 | 4:03 | Stalled |
| 120.2456 | 30.84817 | 30 | 4:03 | In operation |
| 120.2526 | 30.84888 | 30 | 4:04 | In operation |
| 120.2554 | 30.84918 | 30 | 4:05 | In operation |
| 120.2569 | 30.84937 | 0 | 4:05 | Stalled |
| 120.257 | 30.8494 | 10 | 4:06 | In operation |
| 120.2572 | 30.84955 | 16 | 4:06 | In operation |
| 120.2573 | 30.8498 | 14 | 4:06 | In operation |
| 120.2573 | 30.85002 | 6 | 4:07 | In operation |

## 2.2. Problem Description

In this study, waste collection vehicles are the subject of investigation. There is one sanitation base, N is the set of waste collection points, and K is the set of collection vehicles in the collection network. The problem is described in Figure 1. Firstly, the collection network is divided into several sub-areas, and the vehicles start from the sanitation base in an empty state. Each vehicle is responsible for the collection demand of only one sub-area, and the vehicle completes all the collection tasks and returns to the sanitation base under the condition of satisfying the time window constraints and capacity constraints.

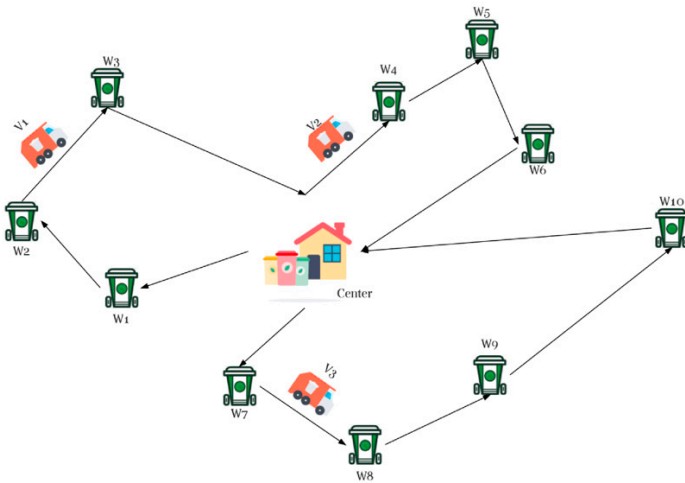

**Figure 1.** An illustration of the problem.

The objective of this study is to find the transportation path that minimizes the total cost. The total cost includes the fixed cost of the vehicle, the transportation cost, and the carbon emissions cost. The relevant assumptions involved in the proposed vehicle routing problem for MSW collection are:

(1) The location of each waste collection point and sanitation base is known, as well as the distances between the nodes.

(2) The amount of waste at each collection point is known and does not exceed the vehicle capacity limit.

(3) A vehicle can only serve one collection route, and the total amount of waste on that route must not exceed the vehicle capacity limit.

(4) All vehicles travel at the same speed and capacity.

(5) There is no traffic congestion, and vehicle speeds are consistent.

### 2.3. Mathematical Model

By analyzing the objectives and constraints affecting the decision-making in waste collection and transportation, a mathematical model for the optimization of a green vehicle route for MSW collection and transportation is established. The meaning of each parameter, set, and decision variable in the model is shown in Table 3:

**Table 3.** Definition of variables and parameters related to the model.

| Element | Description |
| :---: | :---: |
| $N$ | Set of waste collection points, $i = 1, 2, \ldots, N$ |
| $K$ | Set of vehicles, $k = 1, 2, \ldots, K$ |
| $Q$ | Maximum load of the vehicle |
| $d_{ij}$ | Distance between points "$i, j$" |
| $f_k$ | Fixed costs for vehicle $k$ |
| $t_{ik}$ | Time when vehicle $k$ arrives at customer $i$ |
| $c_k$ | Cost per unit distance transported for vehicle $k$ |
| $\alpha$ | Transport costs per unit distance |
| $\beta$ | Time window penalty factor |
| E | Costs of carbon emissions |
| $F_e$ | Fuel consumption per km |
| $\delta$ | Carbon emission factor |
| $q_i$ | Quantity demanded at waste collection point $i$ |
| $[a_i, b_i]$ | Working time window for collection point $i$ |
| $[S_1, S_2]$ | Waste collection time window |
| $x_{ijk}$ | 1 if vehicle $k$ from collection point $i$ to $j$; 0 otherwise |
| $y_{ik}$ | 1 if vehicle $k$ at collection point $i$ for waste collection; 0 otherwise |

In this study, the routes are mainly evaluated in terms of economics. Vehicle fixed costs include vehicle acquisition costs, vehicle maintenance costs, vehicle insurance costs, and staff salaries, as shown in Equation (1). The transportation cost mainly consists of the fuel consumption cost incurred during travel, as shown in Equation (2).

$$f_k = f_p + f_m + f_w + f_i \tag{1}$$

where $f_k$ is the fixed cost of per vehicle, $f_p$ is the acquisition cost of the vehicle, $f_m$ is the maintenance cost of the vehicle, $f_w$ is the cost of vehicle insurance, and $f_i$ is the cost of staff salary.

$$c_k = d_k \times c_f \times \alpha \tag{2}$$

where $c_k$ is the transportation cost of the waste collection vehicle, $d_k$ is the straight-line distance traveled by the vehicle, and $c_f$ is the fuel cost per kilometer. Note that the non-linear coefficient "$\alpha$" is used to describe the actual distance of the vehicle in this paper. The non-linear coefficient is the ratio of the actual traffic distance between the start and end points of a road section to the spatial straight-line distance between the two points.

Vehicle carbon emissions are directly related to fuel consumption and the type of fuel. In order to more accurately measure carbon emissions during vehicle travel, this paper introduces a carbon emission coefficient. The conversion method involves multiplying the vehicle fuel consumption by the corresponding carbon emission coefficient to directly calculate carbon emission. The conversion between fuel consumption and carbon dioxide emission is achieved through a linear relation. This paper measures the costs of carbon emissions by considering the carbon tax paid by enterprises and includes it in the total costs of the target function. The carbon emission costs of the waste collection and transportation vehicles are represented by Equation (3).

$$E = C_m \times \delta \times F_e \tag{3}$$

where $E$ is the cost of carbon emissions, $C_m$ is the carbon tax per unit of carbon emissions, $\delta$ is the $CO_2$ emission coefficient, and $F_e$ is the fuel consumption per km.

The objective function is:

$$\min \sum_{i=0}^{N} \sum_{j=0}^{N} \sum_{k=1}^{K} x_{ijk} d_{ij} c_k + \sum_{j=1}^{N} \sum_{k=1}^{K} x_{0jk} f_k + E \sum_{i=0}^{N} \sum_{j=0}^{N} \sum_{k=1}^{K} x_{ijk} d_{ij} \tag{4}$$

Equation (4) aims to minimize the cost of waste collection paths, which includes fixed costs, transportation costs, and carbon emission costs.

$$\sum_{j=1}^{N} x_{0jk} = 1, \forall k \in K \tag{5}$$

$$\sum_{i=1}^{N} x_{i0k} = 1, \forall k \in K \tag{6}$$

Constraints (5) and (6) indicate that every waste vehicle must depart from the depot and return to the depot after completing the waste collection work.

$$\sum_{j \in N} \sum_{k \in K} x_{ijk} = 1, \forall i \in N, i \neq j \tag{7}$$

Constraint (7) covers the fact that each collection point can only be served by one vehicle and visited once.

$$\sum_{i=1}^{N} x_{ink} - \sum_{j=1}^{N} x_{njk} = 0, \forall n \in N, \forall k \in K \tag{8}$$

Constraint (8) represents the flow balance constraint; that is, the number of vehicles entering a waste collection point is equal to the number of vehicles leaving the waste collection point.

$$\sum_{i \in N} \sum_{j \in N} x_{ijk} \leq N - 1, \forall k \in K \tag{9}$$

Constraint (9) represents the branch circuit elimination constraints and ensures that there are no sub-travels.

$$\sum_{i=1}^{N} \sum_{j=1}^{N} x_{ijk} q_i \leq Q, \forall k \in K \tag{10}$$

Constraint (10) guarantees that the amount of waste collected by each vehicle cannot exceed its maximum load capacity.

$$S_1 \leq t_{ik} \leq S_2, \forall i \in N, \forall k \in K \tag{11}$$

Constraint (11) imposes that the vehicles start and end working times are set within the time windows.

$$a_i \left( \sum_{i=1}^{N} x_{ijk} \right) \leq t_{ik} \leq b_i \left( \sum_{i=1}^{N} x_{ijk} \right), \forall i \in N, \forall k \in K \tag{12}$$

Constraint (12) enforces that the waste must be collected within the time windows.

### 3. Methodology

An improved ant colony algorithm has been designed to assist companies in making decisions. Combining the max–min ant system and the 2-opt local search algorithm to optimize the traditional ant colony algorithm, the solution steps are shown in Figure 2:

Step 1: Initialization of the colony.

Initialize parameters such as colony size, pheromone importance factor, heuristic function importance factor, and maximum number of iterations. Place the ants randomly on different nodes.

Step 2: Constructing the solution space.

Place individual ants in the current solution set, transfer each ant to the next node according to the probability $P_{ij}$, add this node to the current solution set, and repeat this process several times until all ants have visited all nodes. Considering the constraints on vehicle capacity, the state transfer method for ants has been improved, as shown in Equation (13)

$$P_{ij} = \begin{cases} \dfrac{[\tau_{ij}]^{\alpha} [\eta_{ij}]^{\beta} [k_{ij}]^{\lambda}}{\sum_n [\tau_{in}]^{\alpha} [\eta_{in}]^{\beta} [k_{in}]^{\lambda}}, & p \leq p_0 \\ [\tau_{ij}]^{\alpha} [\eta_{ij}]^{\beta} [k_{ij}]^{\lambda}, & p > p_0 \end{cases} \tag{13}$$

where $n$ is the combination of all waste collection points; $\tau_{ij}$ is the pheromone concentration on path $(i, j)$; $\alpha(\alpha \geq 0)$ and $\beta(\beta \geq 0)$ are used for weighting the pheromone intensity and visibility; $\lambda(\lambda \geq 0)$ is the vehicle weight factor; $p$ is a random number in the interval [0,1]; and $p_0$ is a fixed value in the range (0,1);

$\eta_{ij}$ is the visibility on path $(i, j)$. The range of values is (0,1); $\eta_{ij}$ represents the visibility on path $(i, j)$, i.e., the degree of illumination from collection point $i$ to $j$. This is shown in Equation (14):

$$\eta_{ij} = \frac{1}{d_{ij}} \tag{14}$$

where $d_{ij}$ is the distance from collection point $i$ to $j$.

$k_{ij}$ represents the load factor of the collection vehicle and is expressed as Equation (15):

$$k_{ij} = (q_i + q_j) / Q \tag{15}$$

where $q_i$ is the amount of waste at collection point $i$; $Q$ is the vehicle load constraint.

```
                              ╭─────────╮
                              │  Start  │
                              ╰─────────╯
                                   │
                 ┌─────────────────────────────────────┐
                 │ Initial population，Initial number   │
                 │ of iterations                        │
                 │          N_c = 0                     │
                 └─────────────────────────────────────┘
                                   │
                 ┌─────────────────────────────────────┐
                 │ Set the taboo list of each ant       │
                 │ crawling to all nodes                │
                 └─────────────────────────────────────┘
                                   │
                 ┌─────────────────────────────────────┐ ◄──┐
                 │ Number of iterations N_c = N_c + 1   │    │
                 └─────────────────────────────────────┘    │
                                   │                         │
                 ┌─────────────────────────────────────┐    │
                 │            Ant k = 1                 │    │
                 └─────────────────────────────────────┘    │
                                   │                         │
          ┌──►   ┌─────────────────────────────────────┐    │
          │      │          Ant k = k + 1               │    │
          │      └─────────────────────────────────────┘    │
          │                        │                         │
          │      ┌─────────────────────────────────────┐    │
          │      │ Select the next access point         │    │
          │      │ according to the state transition    │    │
          │      │ probability formula                  │    │
          │      └─────────────────────────────────────┘    │
          │                        │                         │
          │      ┌─────────────────────────────────────┐    │
          │      │        Update Taboo Table            │    │
          │      └─────────────────────────────────────┘    │
          │  No                    │                         │
          └──────◄  k ≥ number of ants m                     │
                                   │ Yes                     │
                 ┌─────────────────────────────────────┐    │
                 │    Update pheromone concentration    │    │
                 └─────────────────────────────────────┘    │
                                   │                    No   │
                   Satisfy the termination condition ───────┘
                                   │ Yes
                 ┌─────────────────────────────────────┐
                 │ Get the local optimal solution Local │
                 └─────────────────────────────────────┘
                                   │
                 ┌─────────────────────────────────────┐
                 │ Local optimization of Llocal using   │
                 │ 2-opt and two-point exchange         │
                 └─────────────────────────────────────┘
                                   │
                 ┌─────────────────────────────────────┐
                 │ Output the optimal solution and the  │
                 │ optimal path                         │
                 └─────────────────────────────────────┘
                                   │
                              ╭─────────╮
                              │   End   │
                              ╰─────────╯
```

**Figure 2.** The basic solution process of the improved ACO algorithm.

Step 3: Updating the pheromone.

Calculate the path length $L_k$ visited by each ant and save the optimal solution in the current iteration number. At the same time, update the pheromone concentration on the paths between each node by adding additional pheromone to the feasible paths as well as the optimal paths of the objective function, as shown in Equation (16)

$$\tau_{ij}(t+1) = (1 - \rho) \times \tau_{ij}(t) + \Delta\tau_{ij}^k + \Delta\tau_{ij}^* \tag{16}$$

where $\rho$ denotes the pheromone volatility factor; $(1 - \rho)$ denotes the pheromone residual factor; when the path $(i, j)$ is the feasible path or the path with the minimum objective function, an additional pheromone $\Delta\tau_{ij}^k$ or $\Delta\tau_{ij}^*$ is added to the path $(i, j)$.

Step 4: Premature convergence judgement.

Introduce the max–min ant colony algorithm to overcome the premature convergence problem, keeping the pheromone concentrations after each update within the range of $[\tau_{\min}, \tau_{\max}]$ to prevent large differences in pheromone concentrations between paths. The

minimum concentration of pheromones can increase the likelihood of exploring the optimal solution, while the maximum concentration of pheromones can ensure that ants benefit from past experiences. The equation for the max–min ant colony algorithm is as follows:

$$\tau_{ij} = \begin{cases} \tau_{max}, \; \tau_{ij} > \tau_{max} \\ \frac{\tau_{min} + \tau_{max}}{2}, \; \tau_{ij} < \tau_{min} \end{cases} \tag{17}$$

The clustering degree of a single ant and whether the solution tends to be smooth are used to judge whether the algorithm converges too early. For the aggregation degree of a single ant in an ant colony, the following Equation (18) is used:

$$\sigma^2 = \sum_{i=1}^{n} \left( \frac{f_i - f_{avg}}{f} \right)^2 \tag{18}$$

$$f = \max\{1, \; \max\{|f_i - f_{avg}|\}\} \tag{19}$$

where $\sigma^2$ is the variance of the colony fitness; $f_i$ is the fitness of the $i$th ant; and $f_{avg}$ is the average fitness of the colony. The size of $\sigma^2$ is constrained by taking constant values for $f$. When $\sigma^2 < \sigma^2_{min}$, then it is considered that individuals in the population exhibit aggregation and the algorithm enters into premature convergence. To select some of the better solutions in the initial solution, optimize again; otherwise, continue with the next process.

Step 5: The 2-opt algorithm further optimizes the solution.

Use the 2-opt algorithm to address the issue of the algorithm getting stuck in local optima. The principle is to update the two edges of the exchange solution until the optimal solution is found.

Step 6: Terminating the iteration.

Determine if the maximum number of iterations has been reached and output the optimal result. If the termination condition is not met, repeat Steps 2 to 5.

## 4. Results

This section explores waste collection and transport in Huzhou through a case study. The parameter settings in the model are shown in Table 4.

**Table 4.** Parameterization.

| Element | Description | Value | Unit |
|---------|-------------|-------|------|
| $Q$ | Maximum load of the vehicle | 5 | t |
| $f_k$ | Fixed costs for vehicle $k$ | — | 10,000 RMB |
| $f_p$ | Acquisition costs of the vehicle | 54.9 | CNY/vehicle/day |
| $f_m$ | Maintenance costs of the vehicle | 50 | CNY/vehicle/day |
| $f_w$ | Cost of vehicle insurance | 3.79 | CNY/vehicle/day |
| $f_i$ | Cost of staff salary | 170 | CNY/person/day |
| $c_k$ | Cost per unit distance transported for vehicle $k$ | — | 10,000 CNY |
| $c_f$ | Fuel cost per kilometer | 3 | CNY/km |
| $v$ | Average vehicle travel speed | 30 | km/h |
| $t_i$ | Average operating time at collection points | 0.1 | h |
| $\alpha$ | Non-linear coefficient | 1.4 | — |
| $\beta$ | Road congestion factor | 1.5 | — |
| $F_e$ | Fuel consumption per km | 0.45 | L/km |
| $\delta$ | Carbon emission factor | 3.096 | — |
| $C_m$ | Carbon tax | 0.6 | CNY/kg |

### 4.1. Spatial Clustering of Waste Collection Points

In this paper, the data for longitude, latitude, position, speed, time, and state of collection vehicles at 98 waste collection points in 63 communities of the city are analyzed.

The application of spatial clustering techniques efficiently partitions the 98 waste collection points into five distinct sub-areas (as shown in Figure 3). This subdivision facilitates the strategic deployment of multiple collection vehicles concurrently operating across diverse regions, yielding heightened operational efficiency and substantial cost reductions.

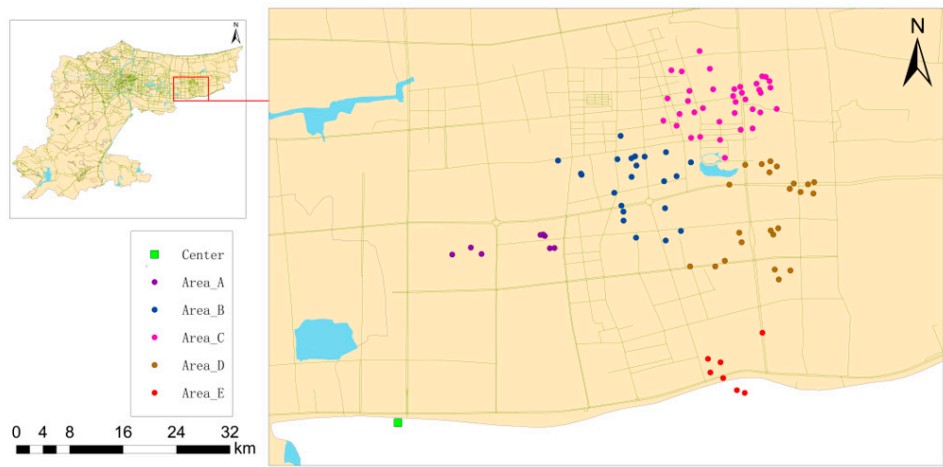

**Figure 3.** Refinement of waste collection point partitioning.

### 4.2. The Optimized Scheme of Collection and Transport Vehicle Dispatching

Utilizing an improved ant colony algorithm, the model was employed to address the task, resulting in the optimal collection and transportation scheme after 200 iterative cycles. The convergence process of the algorithm and the best collecting and transporting path are shown in Figures 4 and 5. Among them, the total distance traveled by vehicles was 98.86 km, the total cost was 2488.91 CNY, and the vehicle configuration was set for five vehicles.

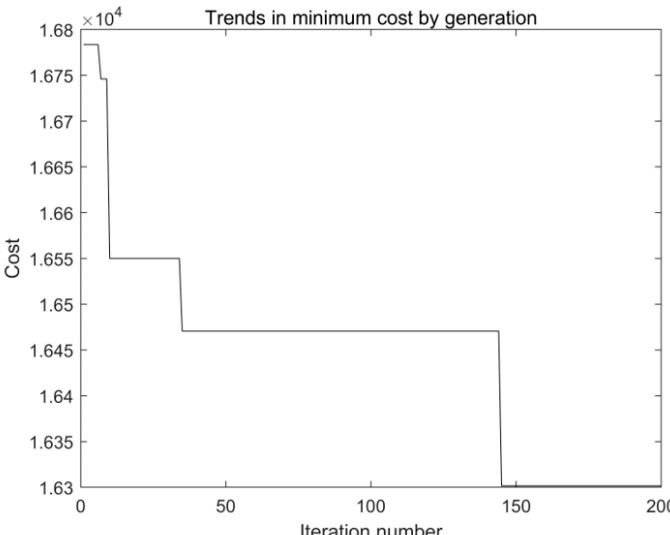

**Figure 4.** Iteration curve of improved ant colony algorithm.

The specific collection and transportation order for each area is presented in Table 5. From the table, it can be seen that the waste collection and transportation work of Area A–E was completed by five vehicles, numbered 1–5. Comprehensively considering the cost of collection and transportation, driving distance, and running time, it was arranged for the No. 1 vehicle to complete the collection and transportation of the eight waste collection points in Area A and then depart from the sanitation base to Area E for collection and transportation. Area C required the No. 3 vehicle and No. 4 vehicle to work at the same time and complete the delivery within the specified working hours.

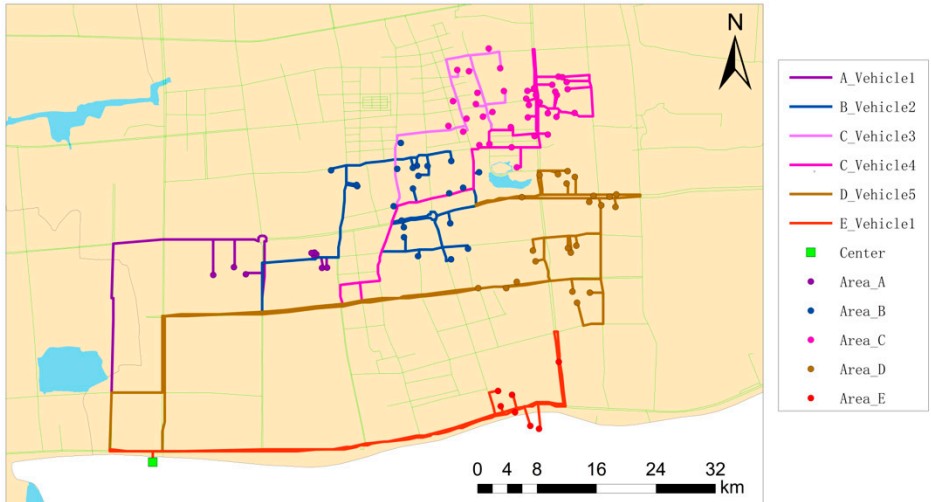

**Figure 5.** Improved route map of each vehicle.

**Table 5.** The sequence of waste collection and transportation within each respective area.

| Area | Vehicle Number | Waste Collection Sequence | Distance Traveled (km) | Assignment Time (h) |
|---|---|---|---|---|
| A | 1 | 0→95→94→93→83→82→81→98→97→0 | 9.43 | [0,1.27] |
| B | 2 | 0→90→91→92→89→78→79→77→87→88 →85→86→76→80→84→15→17→5→20 →19→60→61→96→0 | 17.97 | [0,3.09] |
| C | 3 | 0→22→9→6→1→8→7→3→2→10→13 →14→11→12→0 | 19.65 | [0,2.68] |
| C | 4 | 0→16→18→21→47→41→42→44→45 →43→40→35→36→31→32→37→33→34 →38→39→46→30→29→54→4→0 | 20.21 | [0,3.31] |
| D | 5 | 0→58→68→69→66→67→23→27→28 →25→26→24→53→52→49→51→50→48 →64→65→63→70→72→71→0 | 19.23 | [0,3.26] |
| E | 1 | 0→56→74→62→59→55→73→75→0 | 12.35 | [1.27,2.59] |

*4.3. Scheme Comparison*

In order to test the effectiveness of the method, this study compares the results of the GA and PSO with the transportation scheme, and Tables 6 and 7 show the order of waste collection and transportation for the genetic algorithm and particle swarm algorithm for each area.

As a whole, the schemes of the GA and PSO are somewhat different from the scheme of the improved ant colony algorithm, with the main differences being the order of collection and transportation in the three areas of B, C, and D.

According to the analysis of the actual vehicle track data in Huzhou, the total travel distance of the current waste collection and transportation schemes is 132.34 km. The total travel distance of the vehicle is 111.75 km in the unclustered case and 108.81 km in the genetic algorithm, the total travel distance in the PSO algorithm is 110.04 km, and that in the improved ant colony algorithm is 98.86 km, which is shorter than the distance traveled by the vehicles in all the other schemes.

Each scenario utilizes a comparison between the number of vehicles and the distance traveled, as depicted in Table 8.

**Table 6.** Waste collection and transportation sequence scheme based on GA.

| Area | Vehicle Number | Waste Collection Sequence | Distance Traveled (km) | Assignment Time (h) |
|---|---|---|---|---|
| A | 1 | 0→95→94→93→83→82→81→98→97→0 | 9.43 | [0,1.27] |
| B | 2 | 0→96→90→91→92→89→78→79→77→87 →88→80→76→86→85→84→15→17→5 →20→19→60→61→0 | 21.73 | [0,3.17] |
| C | 3 | 0→12→9→1→6→8→7→3→2→10→13 →14→16→18→21→54→0 | 21.65 | [0,2.73] |
| C | 4 | 0→22→11→47→41→42→44→45→43→40 →35→36→31→32→37→33→34→38→39 →46→30→29→4→0 | 22.70 | [0,3.43] |
| D | 5 | 0→58→68→69→66→67→23→27→28→25 →26→24→52→53→49→50→51→48→64 →65→63→70→72→71→0 | 20.95 | [0,3.37] |
| E | 1 | 0→56→74→62→59→55→73→75→0 | 12.35 | [1.27,2.59] |

**Table 7.** Waste collection and transportation sequence scheme based on PSO.

| Area | Vehicle Number | Waste Collection Sequence | Distance Traveled (km) | Assignment Time (h) |
|---|---|---|---|---|
| A | 1 | 0→95→94→93→83→82→81→98→97→0 | 9.43 | [0,1.27] |
| B | 2 | 0→79→77→78→87→88→80→76→86→85 →84→15→17→5→20→19→60→61→96 →90→91→92→89→0 | 22.38 | [0,3.24] |
| C | 3 | 0→9→1→6→8→7→44→42→41→47→21 →18→16→4→0 | 21.63 | [0,2.78] |
| C | 4 | 0→22→12→11→14→13→10→2→3→45 →43→40→35→36→31→32→37→33→34 →46→39→38→30→29→54→0 | 22.06 | [0,3.46] |
| D | 5 | 0→58→68→69→66→67→23→27→28→25 →24→26→52→53→49→50→51→48→64 →65→63→72→70→71→0 | 22.39 | [0,3.37] |
| E | 1 | 0→56→74→62→59→55→73→75→0 | 12.35 | [1.27,2.59] |

**Table 8.** Comparison of the distance traveled by the collecting and transporting vehicles.

| | Improved ACO (km) | GA (km) | PSO (km) | ACO (km) | Practical Application (km) |
|---|---|---|---|---|---|
| Vehicle_1 | 21.78 | 21.78 | 21.78 | 20.62 | 13.56 |
| Vehicle_2 | 17.97 | 21.73 | 22.38 | 18.87 | 22.47 |
| Vehicle_3 | 19.65 | 21.65 | 21.63 | 21.13 | 14.19 |
| Vehicle_4 | 20.21 | 22.7 | 22.06 | 17.75 | 13.83 |
| Vehicle_5 | 19.23 | 20.95 | 22.39 | 18.92 | 18.89 |
| Vehicle_6 | / | / | / | 18.46 | 17.57 |
| Vehicle_7 | / | / | / | / | 11.7 |
| Vehicle_8 | / | / | / | / | 20.13 |

In contrast, the improved ant colony algorithm is superior to other algorithms in terms of vehicle configuration and driving distance when the vehicle capacity and working time window constraints are met.

A comparison of the costs of each of the waste collection options is shown in Figure 6.

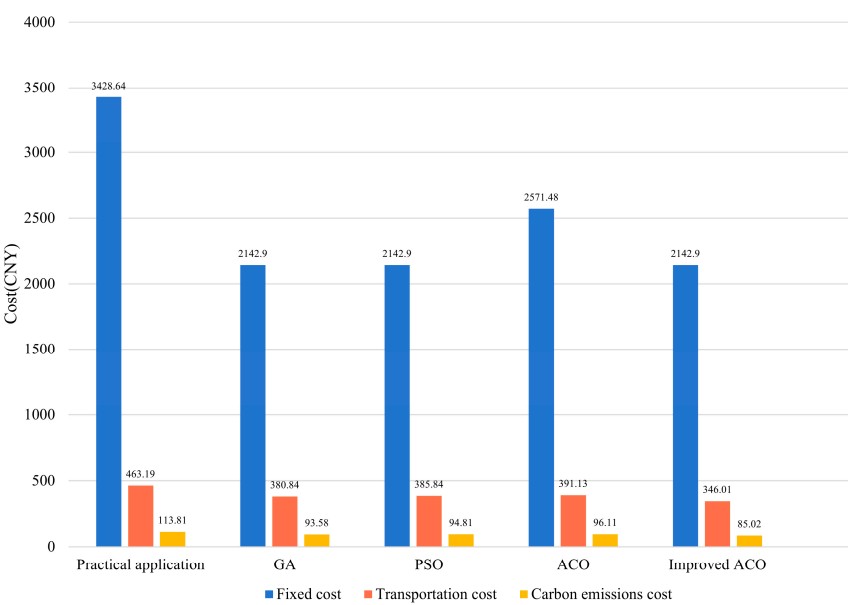

**Figure 6.** Cost comparison of waste collection and transportation solutions.

In summary, the improved ant colony algorithm based on spatial dynamic clustering yields the optimal waste collection and transportation solution with the lowest total cost of 2573.93 CNY. The total cost was reduced by 31.2, 1.66, and 1.89% compared to that of the actual collection and transportation scheme, the GA scheme and the PSO scheme, respectively.

## 5. Conclusions

### 5.1. Results

By addressing the challenges of eco-friendliness and low carbon emissions in vehicle scheduling and route optimization for waste collection and transportation, this study actively aligns with the "carbon neutrality" policy. A model for waste collection and transportation is devised, emphasizing green and low-carbon practices in vehicle routing and navigation. The proposed model is tackled utilizing an enhanced ant colony algorithm.

To confirm the effectiveness and feasibility of the model and algorithm, this research employs various algorithms to address the optimal strategy for MSW collection and transportation in the city. The outcomes indicate the following: (1) The improved ant colony algorithm proposed in this study effectively tackles the model. The improved ACO algorithm is faster and has better performance than the GA and PSO algorithm. Moreover, compared to the GA and PSO collection and transportation schemes, the scheme with the improved ACO algorithm has the shortest total distance traveled and the lowest total cost, which is 1.66 and 1.89% less than that of GA and PSO, respectively. (2) Calculations show that the optimized route reduces total vehicle operating costs by 31.2%, fixed costs by 60%, and carbon emissions by 25.3%, with significant economic benefits.

### 5.2. Discussion

This study still has some deficiencies and directions for further research, as follows:

(1) This study primarily aims to help enterprises optimize their carbon emission portfolio and vehicle routes to maximize environmental and economic benefits. In order to avoid overly complex vehicle routes that would increase the cost of transportation, this study focuses solely on periodic collection point paths. This approach aims to maximize the benefits of garbage collection and transportation while minimizing the number of vehicles required.

(2) The data utilized in this study primarily consist of the factual records of garbage collection and transportation within the designated study area over a period of three consecutive months. In future research, it is recommended to collect a substantial amount

of data that encompass a wider range of temporal and spatial characteristics. This will enable the targeting of vehicle scheduling in advance by predicting seasonal and cyclical changes in garbage collection points.

**Author Contributions:** Conceptualization, S.D. and C.L.; methodology, T.L. and Y.W.; software, C.L. and T.L.; validation, H.L., S.D., and T.L.; formal analysis, S.D.; investigation, C.L. and T.L.; resources, S.D. and Y.W.; data curation, C.L. and H.L.; writing—original draft preparation, C.L., S.D., and T.L.; writing—review and editing, S.D. and T.L.; visualization, S.D., H.L., and Y.W.; supervision, S.D. and Y.W.; project administration, S.D. and T.L.; funding acquisition, S.D. All authors have read and agreed to the published version of the manuscript.

**Funding:** This research was funded by the "Postgraduate Research & Practice Innovation Program of Jiangsu Province", grant number SJCX22_1772; the "Yangzhou Science and Technology Bureau", grant number YZ2021169; the "China Ministry of Education, Humanities and Social Science Fund Project", grant number 22YJAZH139; and the "National Natural Science Foundation of China", grant number 71871035.

**Institutional Review Board Statement:** Not applicable.

**Informed Consent Statement:** Not applicable.

**Data Availability Statement:** The data presented in this study are available upon request from the corresponding author. The data are not publicly available due to privacy.

**Conflicts of Interest:** There is no conflict of interest.

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
