# Peer review of "Optimization of Green Vehicle Paths Considering the Impact of Carbon Emissions: A Case Study of Municipal Solid Waste Collection and Transportation"

_sustainability, doi:10.3390/su152216128_

Round 1

Reviewer 1 Report

Comments and Suggestions for Authors

Line 15-18: The abstract does not convey its intended meaning, and sentences seem incomplete.

Abstracts need to be restructured. What exactly is done through which methodology is not mentioned in the abstract?

Line 25: Specify significant advantages in the abstract. Be specific

How much Energy saving, consumption reduction and emission reduction? Need stats specific?

Need to discuss the dataset. Is the data used in the study open-access data? Clarify the same.

Limited literature review? Only 29

Discuss operational approach used to manage waste?

 municipal solid waste (MSW)? The first latter of each word should be capitalised for abbreviation. Applicable for the complete manuscript.

Figure 2 3 and 5: Not clear

Line 280: typo mistake in Table spelling

Table 5: Proposed algorithm stats should be discussed in the last row of the table. If possible use graphical approach.

Authors suggested that an improved ant colony algorithm is designed. However, in the overall work, the improvement made by the authors is absent. Highlight the same.

Also, if the algorithm is designed, all the aspects should be mentioned (e.g. assumptions, constraints, mathematics, etc.)

Comments on the Quality of English Language

Moderate editing of English language required.

Check and compile the manuscript for typos and grammar.

Author Response

Dear reviewer We are the authors of the paper (Manuscript ID: sustainability-2679119). It was great to receive your revisions on this paper, which helped us a lot in revising it.We have revised the paper according to your suggestions. 

Reviewer 2 Report

Comments and Suggestions for Authors

The article is devoted to the optimization of green transport routes taking into account the impact of carbon emissions. Carbon emissions: A case study of municipal solid waste, collection and transportation.

However, there are some comments regarding the work:

1. The Abstract section must be rewritten to reflect the relevance of the problem being solved and the scientific novelty of the solution obtained. Abbreviations in brackets must be included in the text of the article.

2. Keywords must be adjusted, highlighting special terms that characterize the study.

3. At the end of the Introduction section, it is necessary to define the purpose of the scientific research and present the detailed structure of the article with a presentation of the problems to be solved in the following sections.

4. Figure 2 should be made clearer.

5. Why wasn’t graph theory used to optimize routes?

6. The list of cited sources should include more modern publications on optimization of transport traffic problems, for example,

http://dx.doi.org/10.13140/2.1.1634.4324

https://doi.org/10.3390/math11112586

7. It is necessary to add a Discussion section, where the obtained models and the scientific results obtained, describe their advantages and disadvantages, and also give the limitations of the proposed model for optimizing transport routes.

8. Conclusions must be reduced, structured, highlighting the main scientific and practical results obtained, supporting them with numerical results.

Author Response

Dear reviewer, we are the authors of the paper (Manuscript ID: sustainability-2679119). It was great to receive your revisions on this paper, which helped us a lot in revising it.We have revised the paper according to your suggestions. 

Reviewer 3 Report

Comments and Suggestions for Authors

Paper is very strange composed. Section of this paper is defined as Waste and Recycling, but paper is mainly focused on route choice. There is nothing new in route choice research or conclusions, except the fact it was about waste management.

Comments on the Quality of English Language

English is hard to read and understand. Extensive editing should be done. Sentences are not common for English readers.

Author Response

(The authors gave the same response as above.)

Reviewer 4 Report

Comments and Suggestions for Authors

Green vehicle routing problem (GVRP) has been a hot and significant research topic during the past years. The authors develop a GVRP model for municipal solid waste collection problem. An improved ant colony optimization method is proposed to solve the model. Experiments show the proposed method outperforms GA-based and PSO-based methods. Overall, the paper is well-written, and shows promising results for logistics applications. I have only some minor comments for authors to consider.

1. Page 3, Line 104: The country needs to be added after the city.

2. Section 2.1: "(? = 1,2, … , ?)" and "( ? = 1,2, … ,?)" are redundant.

3. Table 1: The four variables, i.e., entry 1, entry 2, ? and ?, need to be further clarified. For example, What are the kinds of data represented by entry 1 and entry 2? Is ? the number of waste collection points or the set of waste collection points? The similar case is also applied for K.

4. Page 4, Line 143: The first letter of "where" should be uppercase. Please check and revise throughout the paper.

5. Figure 2: The variable k should be italic.

6. Page 7, Line 218: "min" and "max" should not be italic since they are not variables. Please check and revise throughout the paper.

7. Page 7, Line 227: Why the size of the variance is constrained by taking constant values? Please add a further explanation.

8. Page 8, Line 233: When the termination criterion is not satisfied, what should we do? There lacks a description of repeating steps.

9. The data description part is missing.

10. In addition to VRP, arc routing problem (ARP) can also be utilized for waste collection routing. Please add related work in the "Introduction" section. Some recent typical studies include: (1) Waste collection in Southern Italy: Solution of a real-life arc routing problem; (2) Constructive heuristics for the mixed capacity arc routing problem under time restrictions with intermediate facilities; (3) Efficient local search strategies for the mixed capacitated arc routing problems under time restrictions with intermediate facilities; (4) Rich arc routing problem in city logistics: Models and solution algorithms using a fluid queue-based time-dependent travel time representation; (5) Ant colony optimization for the arc routing problem with intermediate facilities under capacity and length restrictions; (6) Heuristic method for a mixed capacitated arc routing problem: A refuse collection application.

Comments on the Quality of English Language

Overall, the English language is fine, but some issues need to be properly tackled. Please see them in my comments.

Author Response

(The authors gave the same response as above.)

Round 2

Reviewer 1 Report

Comments and Suggestions for Authors

Compare mathematical models of all algorithms.

Highlight the statical approach of comparison in abstract section

Comments on the Quality of English Language

Minor editing of English language required

Author Response

Dear reviewer, we are the authors of the paper (Manuscript ID: sustainability-2679119). It was great to receive your revisions on this paper, which helped us a lot in revising it.We have revised the paper according to your suggestions. We have carefully reviewed your comments and have made the appropriate changes in the manuscript based on your suggestions and hope to receive your approval.
Thank you again for giving us the opportunity to resubmit the manuscript and revise it, and we appreciate any help you can provide.

Reviewer 2 Report

Comments and Suggestions for Authors

The authors have revised the article quite significantly. But it would be good to refine the conclusions further. Draw more conclusions and provide numerical characteristics of the results obtained.

Author Response

(The authors gave the same response as above.)

Reviewer 3 Report

Comments and Suggestions for Authors

I have same remarks again.

1. Why is this paper in section called - Waste and Recycling, when main focus of this paper is about route choice (field of traffic research)?

2.  There is nothing new in route choice research, because there are a lot of studies with focus on this topic.

3. In the Discussion section authors stated that they did not consider influence of traffic conditions. Why, when traffic conditions are crucial for travel time?

4. In the Discussion sections, authors also stated that all data is historical, with suggestion that further studies should analyse dynamic vehicle schedule. That is so obsolete, in present route choice models, they are almost all dynamic.

Comments on the Quality of English Language

English should be checked by some native English speaker, or someone who speaks English well.

Author Response

(The authors gave the same response as above.)

Round 3

Reviewer 3 Report

Comments and Suggestions for Authors

Authors stated that this paper was not about traffic route selection, but using traffic route selection in order to improve waste management. Authors also stated that they was not used impact of traffic conditions because there is no traffic cognestion. In addition, authors stated that points for waste collection are fixed.

My question is:

If there are fixed waste collection points without traffic cognestion, why route choice is important?

It is very easy to make route without traffic condition influence, someone who plan route just have to connect closest points.

I will say again, there is nothing novel in traffic management or in waste management.

Author Response

Dear reviewer, we are the authors of the paper (Manuscript ID: sustainability-2679119). It is a pleasure to receive your questions and suggestions about this thesis, which will be of great help to us in our thesis and research.
